# The Removal of Pb^2+^ from Aqueous Solution by Using Navel Orange Peel Biochar Supported Graphene Oxide: Characteristics, Response Surface Methodology, and Mechanism

**DOI:** 10.3390/ijerph19084790

**Published:** 2022-04-15

**Authors:** Zuwen Liu, Shi Yang, Linan Zhang, Jinfeng Zeng, Shuai Tian, Yuan Lin

**Affiliations:** 1School of Civil and Surveying & Mapping Engineering, Jiangxi University of Science and Technology, Ganzhou 341000, China; yangshi2014@126.com; 2Jiangxi Provincial Key Laboratory of Environmental Geotechnology and Engineering Disaster Control, Ganzhou 341000, China; zjf2603@163.com (J.Z.); tians1037@163.com (S.T.); 3School of Live Sciences, Jinggangshan University, Ji’an 343009, China; 4School of Resources and Environmental Engineering, Jiangxi University of Science and Technology, Ganzhou 341000, China; linyuan20060212@163.com

**Keywords:** biochar, graphene oxide, adsorption capacity, response surface methodology, mechanism of adsorption

## Abstract

The value-added utilization of waste resources to synthesize functional materials is important to achieve the environmentally sustainable development. In this paper, the biochar supported graphene oxide (BGO) materials were prepared by using navel orange peel and natural graphite. The optimal adsorption parameters were analyzed by response surface methodology under the conditions of solution pH, adsorbent dosage, and rotating speed. The adsorption isotherm and kinetic model fitting experiments were carried out according to the optimal adsorption parameters, and the mechanism of BGO adsorption of Pb^2+^ was explained using Scanning Electron Microscope (SEM-EDS), X-ray Photoelectron Spectroscopy (XPS), X-ray Diffraction (XRD), and Fourier Transform Infrared Spectroscopy (FTIR). Compared with virgin biochar, the adsorption capacity of Pb^2+^ on biochar supported graphene oxide was significantly increased. The results of response surface methodology optimization design showed that the order of influence on adsorption of Pb^2+^ was solution pH > adsorbent dosage > rotating speed. The optimal conditions were as follows: solution pH was 4.97, rotating speed was 172.97 rpm, and adsorbent dosage was 0.086 g. In the adsorption–desorption experiment, the desorption efficiency ranged from 54.3 to 63.3%. The process of Pb^2+^ adsorption by BGO is spontaneous and endothermic, mainly through electrostatic interaction and surface complexation. It is a heterogeneous adsorption process with heterogeneous surface, including surface adsorption, external liquid film diffusion, and intra-particle diffusion.

## 1. Introduction

Excessive amounts of heavy metals released by the geochemical cycle [1,2] and human production activities (domestic sewage [3], industrial production [4], mining [5,6]) pose a major threat to the ecological environment system and may have a negative impact on the health of humans, plants, and microorganisms [7,8]. At present, there is no effective degradation pathway for heavy metals in the water environment, and the most common method is centralized recovery treatment after adsorption and desorption [9,10,11,12]. Due to the requirements of social environment and sustainable development, exploring high efficiency and environmentally-friendly adsorbents has become the current research hotspot [13,14,15]. The biochar extracted from various solid wastes was widely used in various environmental treatment [16,17,18] and has a strong application prospect to achieve the goal of “treating waste with waste”.

Biochar can be understood as a continuum of pyrolysis products with very complex chemical composition and characteristics [19]. It contains a variety of negatively charged functional groups [20,21], which can fix heavy metal ions on the surface of biochar. However, there are some problems in the virgin biochar materials, such as limited specific surface area, low porosity, and few adsorption sites, which limit its wide application [22,23,24,25]. Therefore, studies enhance the adsorption capacity of biochar by modifying and loading functional materials [26,27]. Graphene oxide (GO) is the main derivative of graphene. It has attracted wide attention because of its large specific surface area and abundant oxygen-containing functional groups [28,29]. Huang et al. (2017) [30] found that the pore volume, specific surface area, and adsorption performance of BGO were significantly increased. Chen et al. (2013) [31] found that graphene oxide chitosan composite material has a high treatment capacity for heavy metal pollutants.

There are abundant navel orange resources in South Jiangxi, which are deeply loved by people. However, the navel orange peels have low economic value and are often discarded, which releases a large amount of CO_2_ and causes environmental pollution [32]. The navel orange peel contains abundant plant fibers and functional groups, which is a good raw material for preparing biochar. Therefore, this study selected lead, a common and typical heavy metal pollutant in the water environment, as the adsorption object, and navel orange peel and flake graphite were used as the raw materials to prepare BGO composites. The optimal adsorption parameters under solution pH, rotating speed, and dosage were analyzed by response surface method. Thermodynamic and kinetic models were used to describe the adsorption process of BGO composites. Combined with Scanning Electron Microscope (SEM-EDS), X-ray Photoelectron Spectroscopy (XPS), X-ray Diffraction (XRD), and Fourier Transform Infrared Spectroscopy (FTIR), the adsorption mechanism of BGO for Pb^2+^ in aqueous solution was described. The results will provide some basic experimental data and material information for the effective treatment of heavy metals in sewage and provide new ideas for the resource reuse of navel orange peel.

## 2. Materials and Methods 

### 2.1. Preparation of BGO Composites

Navel oranges were purchased from Ganzhou fruit wholesale market in Jiangxi Province. The peels were used as the experimental raw material, the graphene oxide was prepared by the modified hummer method [33], and the ratio of flake graphite: KNO_3_:KMnO_4_:H_2_SO_4_ was 1:1.2:6:46 [34]. The mass ratio of biomass to graphene oxide was determined as 20:1 according to the results of preliminary experiments. About of 73 g of biomass and 200 mL of graphene oxide suspension were put into ultrasonic cleaner for 2 h to obtain a uniformly dispersed and stable brown mixture. It was then freeze-dried for 48 h, and then placed in a tube furnace for pyrolysis. We adopted a slow heating and oxygen-limited pyrolysis conditions, heating to target temperature (300 °C, 500 °C, 700 °C) at a rate of 10 °C/min, and then the pyrolysis was carried out for 2 h. It used a N_2_ flow rate of 20 mL/min. After the pyrolysis, the samples were passed through a 100-mesh sieve and put into a sample bag for dry storage. According to the different pyrolysis temperature, the samples of virgin biochar were marked as B300, B500, and B700, the graphene oxide were marked as GO300, GO500, and GO700, and biochar supported graphene oxide were marked as BGO300, BGO500, and BGO700. The adsorbent information were shown shown in Appendix A.

### 2.2. Experimental Methods

#### 2.2.1. Single-Factor Adsorption Experiment

The single-factor adsorption experiment was carried out in the water bath constant temperature shaker. The influence of factors such as solution pH, rotating speed, and adsorbent dosage on the adsorption effect were investigated. BGO composite was put into a 100 mL polyethylene centrifuge tube, and 50 mL of Pb^2+^ solution (100 mg/L) was added. The experimental operating temperature was 25 ± 2 °C, and the shaking speed was set to 150 rpm. In order to achieve adsorption equilibrium, the reaction time was set to 24 h. The pH value was set adjusted to 5.0 ± 0.1 with 0.1 mol/L HNO_3_ and NaOH solution. This experiment adopted the control variable method—when a single factor changes, other reaction conditions remain unchanged. The design scheme was shown in Table 1. The supernatants were collected through 0.45-μm filter membrane. Finally, the concentrations of Pb^2+^ in each sample were tested after acidification with HNO_3_.

#### 2.2.2. Response Surface Experiment

The factor of solution pH, rotating speed, and adsorbent dosage that have a significant impact on the adsorption performance were selected. The experimental design was optimized by Box–Behnken response surface [35]. The central composite design of the adsorption experiment was carried out at three levels of low, medium, and high, respectively. There were 17 groups of experiments, and the central point experiment was repeated for 5 groups. The experiment was repeated three times in each group, and the average value was taken as the corresponding response value.

#### 2.2.3. Adsorption Isotherm and Kinetic Fitting Experiment

The single-molecular-layer Langmuir model, multi-molecular-layer Freundlich model, and diffusion interface Temkin model were used to perform fitting analysis on the adsorption of Pb^2+^ by BGO700. The Langmuir model can better represent the experimental results when the adsorption on solid surface is quite uniform, and the adsorption is limited to single molecular layer. The Freundlich model and Temkin model can describe the heterogeneous adsorption process of heterogeneous surfaces. The sample was placed into a 100 mL centrifuge tube, then 50 mL of Pb^2+^ solution with different initial concentration (50, 100, 150, 200, 250, 300, 350, 400, 500 mg/L) was added into each of them. The mixtures were shaken at 25, 35, and 45 °C for 24 h in the water bath constant temperature shaker. The pH of the solution and rotating speed are adjusted to 5 and 173 rpm, respectively, according to the optimal experiment conditions coupled with the response surface. The supernatants were collected through 0.45-μm filter membrane. Finally, the concentrations of Pb^2+^ in each sample were tested after acidification with HNO_3_.

The adsorption kinetics of BGO700 and B700 were fitted with quasi-first-order kinetics, quasi-second-order kinetics, and intra-particle diffusion models. The solution of Pb^2+^ with concentration of 100 mg/L was placed in 250 mL conical flask, then shaken in the water bath constant temperature shaker at 25 °C. The adsorbent dosage, solution pH, and rotating speed were adjusted to 0.086 g, 5 and 173 r/min. The water sample was taken out at 1, 5, 10, 30, 60, 90, 120, 240, 360, 600, 900, 1260, 1440, 2160, and 2880 min, respectively. The supernatants were collected through 0.45-μm filter membrane. Finally, the concentrations of Pb^2+^ in each sample was tested after acidification with HNO_3_.

#### 2.2.4. Batch Adsorption–Desorption Experiments

Adsorption experiment was first tested to get saturated adsorbent. An amount of 0.086 g BGO700 adsorbent was added to 100 mL of 100 mg/L Pb^2+^ solution in conical flask. The mixed solution was stirred in a water-bath constant temperature shaker for 24 h and then the pollutant concentration was measured. The saturated adsorbent was collected by vacuum suction filtration. In the desorption experiment, 0.5 mol/L HCl solution was selected as the desorbent [36,37]. The saturated adsorbent was put into 0.5 mol/L HCl solution, the mixed solution was stirred for 12 h, and then the pollutant concentration was measured. The adsorbent of desorption experiment was rinsed with deionized water for several times until the pH of the filtrate was close to neutral. Next, the samples were dried in two different conditions, into the vacuum oven and under an ultraviolet light. The temperature of the vacuum oven was set to 60 °C for 4 h. The ultraviolet drying adopted 256 nm, intensity of 80 W, and height of 10 cm to illuminate for 4 h. The adsorption and desorption experiments were repeated for five times in the same way. There are four groups in this experiment, which are marked as UV/BGO700, UV/B700, Bake/BGO700, Bake/B700, respectively.

### 2.3. Analytical Instruments and Methods

The pH value of solution was tested in the potential method by using a pH meter (Sartorius, PB-10, Göttingen, Germany). The CHNS content of material was analyzed by an element analyzer (EA, vario El cube, Hanau, Germany), and the O element content was calculated by difference method. The microscopic morphology and surface composition of the sample were characterized by a field emission scanning electron microscope (FEI, MLA650F, Hillsboro, OR, USA) with an energy depressive spectroscopy (Bruker, Karlsruhe, Germany). The surface area and pore volume of the samples were determined by Brenner Teller (ASAP, 2020HD88, Norcross, GA, USA). The surface functional groups of the samples were characterized by Fourier Transform Infrared Spectroscopy (NICOLET5700, Waltham, MA, USA) with the wave number range of 4000~400 cm^−1^. The changes of crystal structure and functional groups before and after adsorption were analyzed by X-ray diffraction (DX-2700, Dandong, Liaoning, China) and X-ray photoelectron spectroscopy (Thermo Scientific K-Alpha+, Waltham, MA, USA). The concentrations of Pb^2+^ in solution were determined by using ICP-OES (AvioTM200, Fremont, CA, USA).

### 2.4. Quality Control

In this study, to ensure the quality of the sample analysis, before measuring the concentrations of Pb in solution, a set of Pb standard solution of different concentrations (0, 1.0, 2.0, 5.0, 10, 20, 50 mg/L) was used to correct the ICP-OES and obtain a calibration curve (R^2^ > 0.999). All samples were tested three times, and the error of the parallel determination results was within 10%. 

### 2.5. Data Processing and Analysis

The thermodynamic parameters of adsorption mainly include Gibbs free energy (ΔG), enthalpy change (ΔH), and entropy change (ΔS).

The thermodynamic parameters of adsorption can be calculated by the following equation,
∆G = −R·T·ln(K_L_)(1)
∆G = ∆H − T∆S(2)

From Equations (1) and (2), the following Equation (3) can be obtained:ln(K_L_)= ∆S/R − ∆H/(R·T)(3)
where R stands for general gas constant, 8.314 J/(mol·K); T stands for thermodynamic temperature, K; K_L_ stands for solid–liquid partition coefficient, the ratio of the concentration of Pb^2+^ at adsorption equilibrium (Qe) by BGO700 to the concentration of Pb^2+^ in the solution (Ce), is linearly fitted by taking Qe as the abscissa and ln(Qe/Ce) as the ordinate, and the intercept is ln(K_L_).

## 3. Results and Discussion

### 3.1. Characterization of Biochar Composites

#### 3.1.1. The Analysis of Physical Properties

The physical properties and specific surface area analysis of biochar composites is shown in Table 2. The pH and C content are positively correlated with the pyrolysis temperature, while the H and O contents were decreased with the increase of pyrolysis temperature. The oxygen content was increased after loading graphene oxide, which indicates that the surface oxygen containing functional groups were increased [38]. The atomic ratios H/C, O/C, and (O + N)/C of the composites were decreased with the increase of the pyrolysis temperature, which indicates that the higher degree of carbonization and more complete π-conjugated aromatic structure with high pyrolysis temperature [39]. The specific surface area was increased, which indicates that the pyrolysis temperature is the key factor affecting the specific surface area.

#### 3.1.2. Result of SEM-EDS

The surface micromorphology and component of B700, GO700, and BGO700 is shown in Figure 1. It can be found from Figure 1(a1,a2) that biochar is porous and has abundant specific surface area, and contains some mineral elements of Ca, Mg. It can be found from Figure 1(b1) that GO is a transparent flocculent structure with smooth surface, curly edge, and fold, and has good flexibility. The surface components mainly contain C and O elements, and the content of O elements is relatively high. The GO supported on biochar was shown Figure 1(c1), and there are smooth and folded flakes on the surface. At the same time, the percentage of O element content increased (Table 2), which indicates that the GO was successfully supported on the biochar.

#### 3.1.3. Result of FT-IR

The FT-IR analysis of B700, GO700, and BGO700 is shown in Figure 2. It can be seen from the figure that the peak at 3434 cm^−1^ is the characteristic peak of −OH group [40]. The peak of biochar at 2922.8 cm^−1^ is caused by the stretching vibration of aliphatic hydrocarbons or cycloalkanes −CH_3_ and −CH_2_. After being compounded with GO, the composite material also showed a peak at 2357.49 cm^−1^, where it is unique to GO, and indicates that GO was successfully loaded on biochar. The wavelength around 1620.7 cm^−1^ is the peak of −COOH group [41]. Moreover, before 1700 cm^−1^ the composite material showed alkyl, aromatic, and some oxygen-containing groups. These functional groups can provide adsorption sites and enhance the adsorption performance of the composite material.

### 3.2. Single-Factor Adsorption Experiment of Material

#### 3.2.1. Adsorption Capacity of the Material

The adsorption capacity of Pb^2+^ by different pyrolysis temperatures was shown in Figure 3. It can be seen from the figure that the pyrolysis temperature has a greater influence on the adsorption capacity of the material. The adsorption capacity of BGO700 was increased by 134.9% compared to BGO300, and B700 was increased by 187.1% compared to B300. This is due to the influence of pyrolysis temperature, which is consistent with the results of other studies [42,43]. In addition, orange peel biochar showed the worst adsorption capacity (B300: 21.7 mg/g; B500: 43.43 mg/g; B700: 62.3 mg/g), which was significantly improved by about 97%, 78%, and 62% when biochar was supported by the GO (BGO300: 42.82 mg/g; BGO500: 77.46 mg/g; BGO700: 100.61 mg/g). This indicates that GO has strong adsorption performance and can effectively improve the ability of the navel orange peel biochar for heavy metal ions, so that biomass resources can be fully utilized. 

#### 3.2.2. The Effect of Initial pH on Adsorption Capability

The pH of the solution is an important factor affecting the adsorption of Pb^2+^, and the electrostatic interaction with Pb^2+^ is affected by changing the surface charge of BGO. The effect of different pH on the adsorption of Pb^2+^ by BGO was shown in Figure 4a, which indicated the pH was 6 as the best condition for the adsorption of Pb^2+^. With the decrease of pH, the adsorption capacity of Pb^2+^ decreased, showing the lowest value at pH was 2. This is due to the high concentration of H^+^ in the solution under low pH conditions, which will compete with Pb^2+^. In addition, the radius of hydrated H^+^ ion is smaller and easily adsorbed than the ion Pb^2+^, thus competing for the adsorption site of BGO and inhibiting the adsorption of Pb^2+^ [44]. With the increase of solution pH, H^+^ ion decreased and the negative charge on the BGO surface increased, together with the electrostatic attraction between Pb^2+^ and BGO charged sites.

The morphological distribution of Pb^2+^ in aqueous solution under different pH conditions is analyzed by visual MINTEQ software. The results are shown in Figure 4b. It can be seen from Figure 4b that when the pH value of the solution is less than 6, Pb exists in ionic state, and with the increase of pH, ionic Pb hydrolyzes to form a variety of hydroxyl complexes. Therefore, when the pH value in the solution is high, Pb^2+^ is prone to hydrolyze and form Pb(OH)_2_ precipitation, which is beneficial to the removal of Pb^2+^ from pollutants. At the same time, the formation of hydroxyl complex will also reduce the average charge of Pb^2+^, resulting in a significant decrease in the secondary solvation energy, which is more beneficial to the adsorption of Pb^2+^ on the BGO surface through electrostatic attraction [45]. 

#### 3.2.3. The Effect of Rotating Speed on Adsorption Capability

The influence of different rotating speed conditions on the adsorption of Pb^2+^ by BGO is shown in Figure 4c. It can be seen from Figure 4c that the efficiency of Pb^2+^ adsorption was slightly reduced under low and high-speed conditions. This may be that the diffusion rate of Pb^2+^ increased with the increase of rotational speed, which leads to an increase in the effective collision probability of the adsorbent BGO and Pb^2+^ [46]. When the rotating speed is low, the diffusion rate of Pb^2+^ is low, and the effective collision probability between adsorbent BGO and Pb^2+^ is low. When the rotating speed was increased to 200 rpm, additional hydrodynamic shear stress will be generated on the surface of BGO, which will destroy the bond bridge between BGO particles and Pb^2+^ in the gap, increasing the possibility of desorption of BGO adsorbing Pb^2+^, and reducing the adsorption efficiency of BGO for Pb^2+^. 

#### 3.2.4. The Effect of Adsorbent Dosage on Adsorption Capability

The dosage of adsorbent can determine the adsorption capacity of BGO for Pb^2+^ at a certain concentration, thereby affecting the adsorption efficiency of BGO for Pb^2+^ in water. The adsorption capacity of Pb^2+^ with different dosages of BGO is shown in Figure 4d. The results showed that with increasing of BGO dosage, the adsorption capacity of BGO first increases and then decreases. The reason for the increase may be that the dosage of BGO adsorbent increases, and it is relatively easier to be adsorbed by active sites in the collision process, and the number of adsorption sites is also more. When the content of adsorbent is low, Pb^2+^ is easy to be separated from the surface of the adsorbent under hydraulic shear. When the content of adsorbent is excessive, the adsorbents are easy to overlap with each other, but the effective active sites cannot be fully utilized.

### 3.3. Simulation and Optimization of Pb^2+^ Adsorption Process by Response Surface Methodology

Based on the results of single-factor experiments and environmental factors in engineering practice [47,48], the Box–Behnken method was used to explore the influence of solution pH, rotating speed, and adsorbent dosage on the adsorption capacity of Pb^2+^ by BGO700 in the coupling system. The Pb^2+^ adsorption capacity of BGO700 was used as the response value to explain the adsorption effect. The experimental design and results were shown in Appendix A.

In order to analyze the interaction between the factors and the response value in the experiment, the three-dimensional response surfaces were made according to the regression model equation. We compare the effect of rotating speed and adsorbent dosage against solution pH in Figure 5. The interaction effect between solution pH and rotation speed on Pb^2+^ adsorption capacity of BGO700 under the central value of adsorbent dosage is shown in Figure 5a. The solution pH has a significant effect on the adsorption of BGO700, while the speed has a small effect. The interaction effect between the solution pH and adsorbent dosage on Pb^2+^ adsorption capacity of BGO700 under the central value of rotation speed is shown in Figure 5b. The results showed that the dosage of adsorbent has a certain effect on the adsorption capacity of BGO700, but the effect was not significant. The interaction effect between rotation speed and adsorbent dosage on Pb^2+^ adsorption capacity of BGO700 under the central value of solution pH is shown in Figure 5c. The flat curvature shows that the effect of rotating speed and adsorbent dosage are negligible in comparison to the solution pH, but the influence of adsorbent dosage was slightly higher than the rotation speed. Therefore, the order of the degree of influence on the adsorption of Pb^2+^ was solution pH > adsorbent dosage > rotation speed. The analysis with the quadratic multiple regression model showed that the optimal conditions of obtaining the maximum Pb^2+^ adsorption capacity (125.165 mg/g) were: solution pH was 4.97, rotation speed was 172.97 r/min, adsorbent dosage was 0.086 g.

Considering the feasibility of the actual operation, the solution pH, rotation speed, and adsorbent dosage were set to 5, 173 r/min, and 0.086 g, respectively. According to the optimal conditions, three groups of parallel experiments were carried out on the adsorption of Pb^2+^. The experimental results showed that the adsorption capacity of Pb^2+^ was 118.32 mg/g, and the deviation from the predicted response value was only 5.47%, which indicates that the error between the actual experimental value and predicted value of model was small. At the same time, combined with the relationship between the actual value and the predicted value of the model, Figure 5d further illustrates that the model was more accurate in predicting the influence factors, and can better optimize the conditions of experiment.

### 3.4. Adsorption Thermodynamics and Kinetics after Optimization

The adsorption isotherm is a curve that describes the relationship between the dynamic equilibrium process and parameters of the adsorbent in the solution, which is of great significance for evaluating the practicality of adsorption. At present, isothermal models are commonly used in solid–liquid phase adsorption, including Langmuir, Freundlich, and Temkin [18]. The adsorption isotherms of Pb^2+^ by BGO700 at different temperatures are shown in Figure 6a. When the initial concentration of Pb^2+^ is the same, the adsorption capacity of Pb^2+^ by BGO700 is higher at the highest temperature. This is due to the enhanced thermal and diffusion movement of the molecules, which speeds up the movement rate of adsorbent active site and adsorbate.

The fitting results of Langmuir, Freundlich, and Temkin models are shown in Figure 6b–d, respectively. The relevant fitting parameters calculated by the three models are shown in Table 3. It can be seen from the table that the relative coefficient R^2^ of Langmuir model was less than 0.9, and the theoretical maximum adsorption capacity of Pb^2+^ fitted by Langmuir model was lower than the actual experimental value. The results showed that there is an interaction between Pb^2+^ in the solution, the adsorption process was not single molecule adsorption on uniform surface, and BGO700 adsorption process was not suitable for Langmuir model fitting. The Freundlich model and Temkin model can describe the heterogeneous adsorption process of heterogeneous surfaces. The value of 1/n in Freundlich model was between 0.079 and 0.108, which indicates that the adsorption process is easy to take place and multi-layer adsorption can occur. Therefore, the Freundlich model is more suitable for describing the adsorption process.

The adsorption thermodynamic equation and parameters were used to describe the state characteristics of the adsorption process. According to the experimental results of the isotherm adsorption of Pb^2+^ by BGO700, the linear equation of ln(K_L_)^−1^/T was fitted with 1/T as the abscissa and ln(K_L_) as the ordinate. The parameter fitting results are shown in Figure 7, and the correlation coefficient R^2^ = 0.989, which has a better linear relationship.

The calculation results of adsorption thermodynamic parameters are shown in Table 4. The calculated values of ∆S > 0, which indicates that the process of adsorption by BGO700 is an entropy increasing process, which may be due to the degree of confusion was increased in the reaction system after the addition of composite materials. The calculated values of ∆G at different temperatures (25 °C, 35 °C, and 45 °C) are negative, ranging from −3.934 to −5.094 kJ/mol, indicating that the adsorption process is spontaneous and endothermic. The absolute value of ∆G was increased with the increase of temperature, indicating that increasing the temperature is beneficial to the progress of adsorption.

### 3.5. Optimized Adsorption Kinetics Fitting

The variation of Pb^2+^ adsorption capacity with oscillation time is shown in Figure 8a, from which it can be seen that the adsorption process is divided into three stages: fast adsorption, slow adsorption, and adsorption equilibrium stage. At the initial stage of the reaction, there are many adsorption sites and surface functional groups on the surface of the adsorbent. With the progress of the adsorption reaction, the effective adsorption sites were gradually decreased. Meanwhile, the concentration of Pb^2+^ in the solution gradually decreases, and the mass transfer rate slows down. At slow adsorption stage, the adsorption rate depends on the speed at which Pb^2+^ enters the internal site from the outside of the adsorbent [49]. At the last stage, the solid–liquid two–phase reaches dynamic adsorption equilibrium.

The fitting curves of quasi first–order kinetics, quasi second–order kinetics, and intra-particle diffusion model are shown in Figure 8b–d, and the fitting parameters are shown in Table 5. The quasi second–order kinetics equation has a better fitting effect on the Pb^2+^ adsorption, and the fitting degree reached R^2^ ≥ 0.998. At the same time, the theoretical equilibrium adsorption capacity fitted by quasi second order–kinetic equation was 128.53 mg/g and 81.9 mg/g, which are very similar to the experimental equilibrium adsorption capacity, while the adsorption capacity was fitted by the quasi first–order kinetic equation which was quite different from the experimental value. This indicates that the adsorption process includes surface adsorption, external liquid film diffusion, and intra-particle diffusion. In addition, the intra–particle diffusion model was used to analyze the diffusion rate. As shown in Figure 8d, the adsorption process of Pb^2+^ in solution can be divided into two parts. The first stage is the migration of Pb^2+^ from solution to sample surface, which is controlled by molecular diffusion and membrane diffusion. The second stage may be caused by diffusion within the particles. The result indicates that membrane diffusion and intra–particle diffusion may occur simultaneously, which are the main rate control steps in the whole adsorption process.

### 3.6. Adsorption/Desorption Recycling Experiment

The adsorption/desorption cycle experiment can not only understand the utilization efficiency of the sample but can also help to further understand the adsorption mechanism. The adsorption/desorption capacity of sample for each number of experiments is shown in Figure 9. The adsorption capacity of UV/BGO700, bake/BGO700, UV/B700, and bake/B700 gradually decreased and finally reached a stable trend, which was 57.4 mg/g, 50.6 mg/g, 26.6 mg/g, and 28.5 mg/g, respectively. The reason for the low adsorption efficiency may be that the Pb^2+^, through the action of hydrogen bonds, π–π bonds, and oxygen–containing functional groups, was not completely desorbed by HCl. In addition, the adsorption capacity of the sample irradiated by ultraviolet light is slightly higher than that of the vacuum-dried sample, which is due to the introduction of some oxygen-containing functional groups on the surface of the sample by ultraviolet light irradiation [50]. According to Figure 9b, the desorption efficiency ranged from 54.3 to 63.3%.

### 3.7. Adsorption Mechanism of Pb^2+^ on Biochar-Supported Graphene Oxide

#### 3.7.1. SEM-EDS Analysis after Adsorption

The structural characteristics (specific surface area, micropores) and surface properties (oxygen-containing functional groups) affect the adsorption process. At the same time, various interactions such as surface precipitation, electrostatic interaction, and π–π interaction are also conducive to the adsorption of Pb^2+^. The microscopic morphology and surface composition analysis after adsorption are shown in Figure 10. There are many small spots on the surface and pores of BGO700. According to surface composition analysis, these small spots are adsorbed Pb^2+^. This is because there are abundant oxygen-containing functional groups with negative charge on the surface of BGO700, which can be complex with Pb^2+^. At the same time, the more negative surface charges, the stronger the electrostatic interaction between Pb^2+^ and BGO700. 

#### 3.7.2. XPS Analysis before and after Adsorption

The XPS spectra of total elements and peaks before and after adsorption of Pb^2+^ by BGO700 is shown in Figure 11. It can be seen from Figure 11 that the Pb peaks were added in the total spectrum after adsorption, which are pb4f_5/2_ and pb4f_7/2_, respectively. The corresponding binding energies are 144.23 EV and 139.4 EV detected by XPS. According to the chemical state database of XPS, Pb exists in the form of Pb^2+^ on the surface of BGO700. The peaks of C1s, O1s, and N1s did not shift significantly before and after adsorption, which indicates that the chemical valence states of C, O, N elements and outer electron density did not change basically before and after adsorption of Pb^2+^ by BGO700.

#### 3.7.3. FTIR and XRD Analysis before and after Adsorption

The analysis of FTIR and XRD before and after the adsorption of Pb^2+^ by BGO700 is shown in Figure 12. The oxygen–containing functional groups of BGO700 before the wavelength of 1700 cm^−1^ have changed after adsorption of Pb^2+^, in which the wavelength of C-O group has shifted from 1395.5 cm^−1^ to the left by 26.9 cm^−1^, and the absorption peak of C=O group has shifted from 1126.9 cm^−1^ to 1162.3 cm^−1^. The strength of the stretching vibration was significantly weakened, which indicates that the oxygen–containing functional groups participate in the adsorption process, and the surface complexation with Pb^2+^ occurred on the surface of BGO700.

The XRD diffraction pattern of the sample before and after adsorption is shown in Figure 12b. The broad diffraction peak appears near 2θ = 23.06°, which corresponds to the graphitic carbon structure. The broadened diffraction peaks indicates that the carbon structure is amorphous [51]. This indicates that the graphite crystal structure of the biomass was destroyed during the pyrolysis process, so the biochar composite material has an amorphous structure. In addition, a relatively weak diffraction peak appears at 2θ = 43.18°, which is a characteristic peak of the carbon structure. It is formed by the cracking of the graphene structure during the pyrolysis process [52]. The strong and narrow diffraction peaks were observed in the vicinity of 2θ = 24.32°, 26.82°, 33.66°, 39.86°, 42.36°, which indicates that the adsorption of Pb^2+^ exists as a crystal structure, and that biochar composites could adsorb Pb^2+^ by precipitation.

## 4. Conclusions

Graphene oxide significantly improves the ability of biochar to adsorb heavy metal ions. The pH of the solution has the greatest impact on the adsorption effect of biochar composite material. When the pH of the solution was less than 4, it inhibited the adsorption of Pb^2+^ by BGO. The rotational speed and the initial concentration of Pb^2+^ mainly affect the adsorption performance by affecting the contact rate between Pb^2+^ and the adsorbent surface. The response surface method optimization design experiment results showed that the order of influence on the adsorption of Pb^2+^ was solution pH > adsorbent dosage > rotating speed. The optimum conditions for the maximum Pb^2+^ adsorption capacity (125.17 mg/g) were as follows: solution pH was 4.97, rotation speed was 172.97 rpm, adsorbent dosage was 0.086 g. In the adsorption–desorption experiment, the desorption efficiency ranged from 54.3 to 63.3%. The process of Pb^2+^ adsorption by BGO was spontaneous and endothermic, mainly through electrostatic interaction and surface complexation. It was a heterogeneous adsorption process with heterogeneous surface, including surface adsorption, external liquid film diffusion, and intra-particle diffusion.

## Figures and Tables

**Figure 1 ijerph-19-04790-f001:**
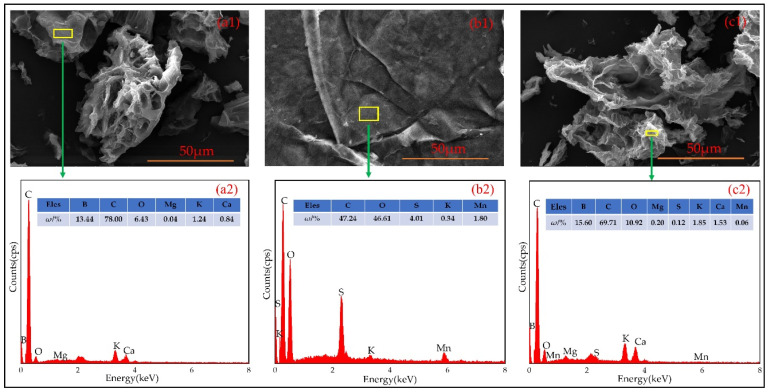
(**a1**) surface micromorphology of B700, (**a2**) component spectra of B700; (**b1**) surface micromorphology of GO700, (**b2**) component spectra of GO700; (**c1**) surface micromorphology of BGO700, (**c2**) component spectra of BGO700. Yellow square in the images indicates the zone where the EDS spectra was performed.

**Figure 2 ijerph-19-04790-f002:**
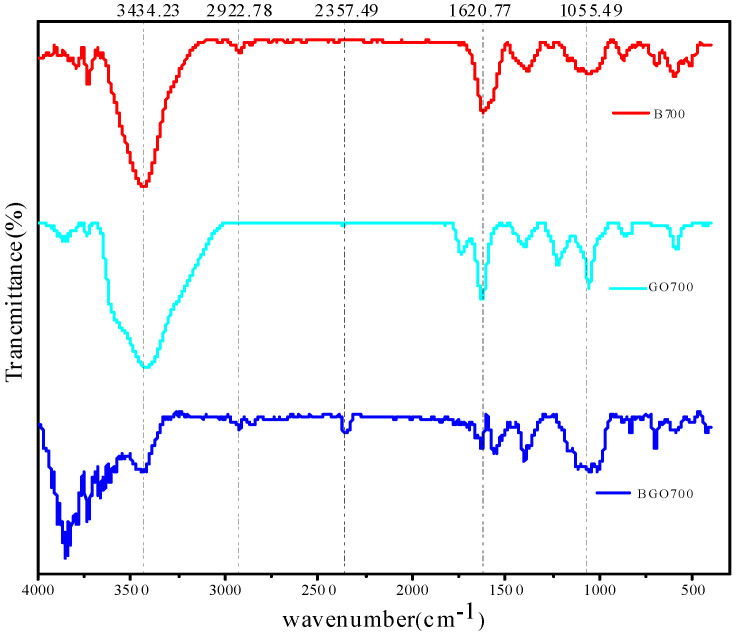
FTIR spectra of the obtained materials: B = biochar, GO = graphene oxide, BGO = composite biochar-graphene oxide.

**Figure 3 ijerph-19-04790-f003:**
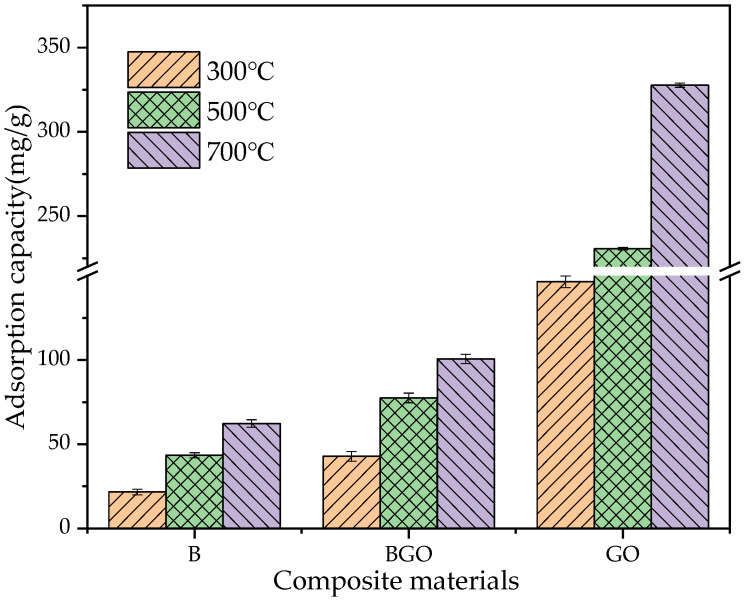
Adsorption capacity of biochar (B), composite biochar-graphene oxide (BGO), and graphene oxide (GO) obtained at different pyrolyzed temperatures.

**Figure 4 ijerph-19-04790-f004:**
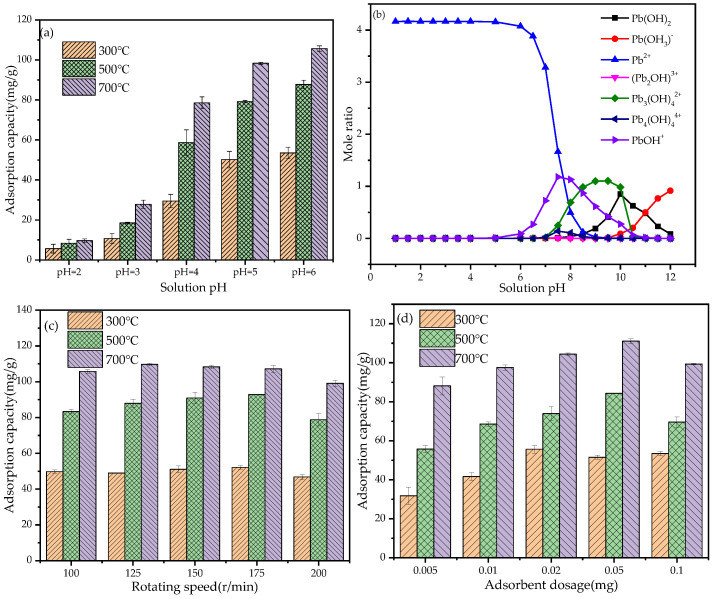
(**a**) The effect of solution pH; (**b**) The morphological distribution of Pb^2+^ in aqueous solution; (**c**) The effect of rotating speed; (**d**) The effect of adsorbent dosage.

**Figure 5 ijerph-19-04790-f005:**
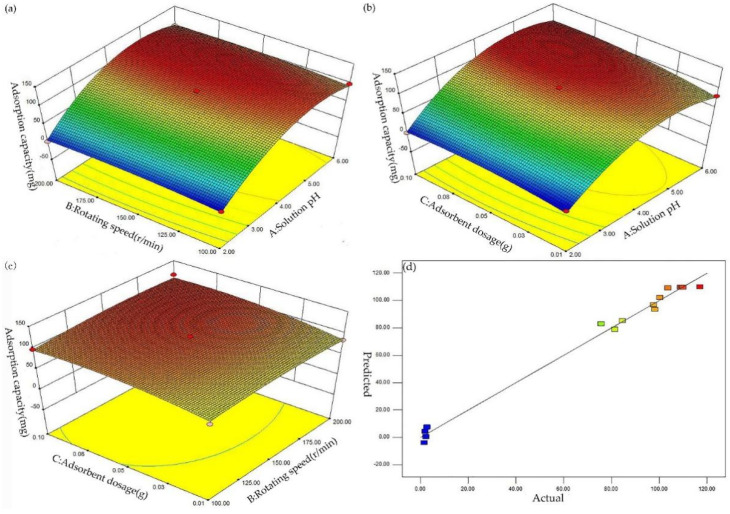
The three–dimensional response surfaces of interaction effect; (**a**) Coupling of rotational speed and pH; (**b**) Coupling of dosage and pH; (**c**) Coupling of dosage and rotational speed; (**d**) Actual value and predicted value.

**Figure 6 ijerph-19-04790-f006:**
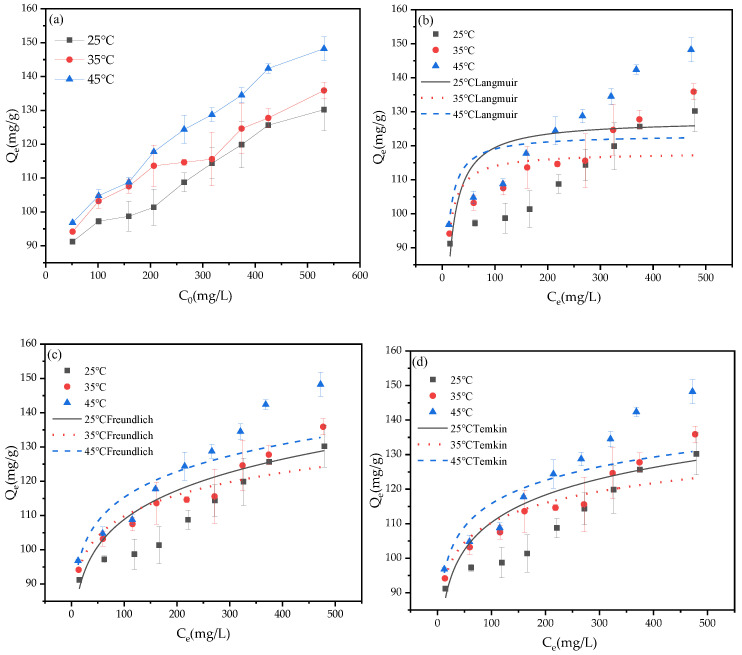
(**a**) The adsorption isotherms of Pb^2+^ by BGO700; (**b**) The adsorption isotherm model of Langmuir; (**c**) The adsorption isotherm model of Freundlich; (**d**) The adsorption isotherm model of Temkin.

**Figure 7 ijerph-19-04790-f007:**
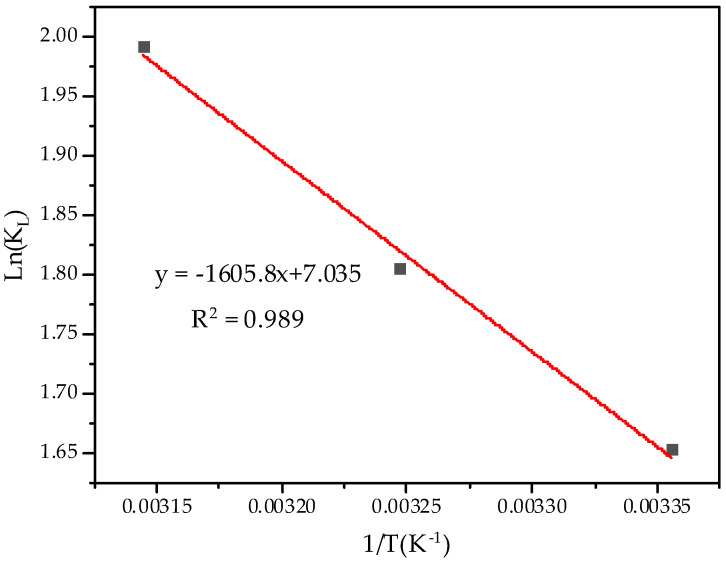
The thermodynamic parameter fitting.

**Figure 8 ijerph-19-04790-f008:**
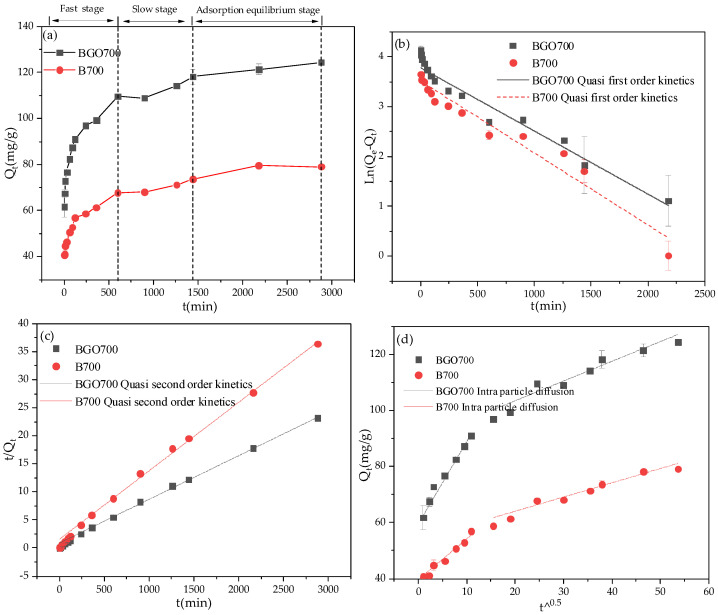
The kinetic fitting curve; (**a**) The variation of Pb^2+^ adsorption capacity with oscillation time; (**b**) The quasi first–order kinetics; (**c**) The quasi second–order kinetics; (**d**) The intra–particle diffusion model.

**Figure 9 ijerph-19-04790-f009:**
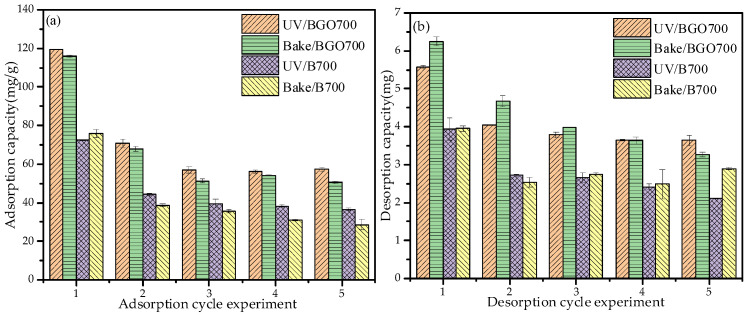
(**a**) The adsorption capacity of cycle experiments; (**b**) The desorption capacity of cycle experiments.

**Figure 10 ijerph-19-04790-f010:**
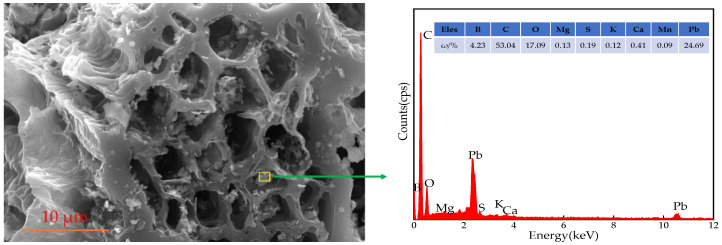
The SEM-EDS images after adsorption.

**Figure 11 ijerph-19-04790-f011:**
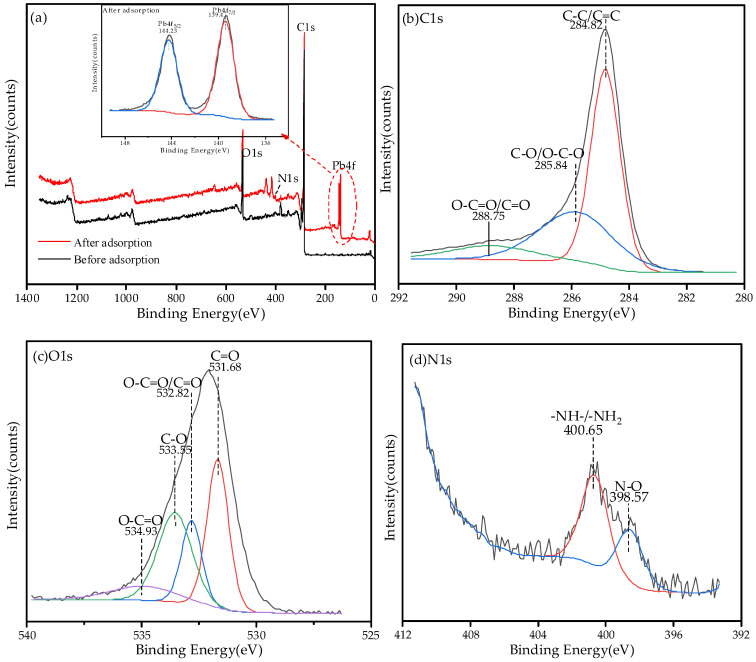
The XPS analysis spectrum before and after adsorption: (**a**) total element, (**b**) C1s peak split, (**c**) O1s peak split, and (**d**) N1s peak split.

**Figure 12 ijerph-19-04790-f012:**
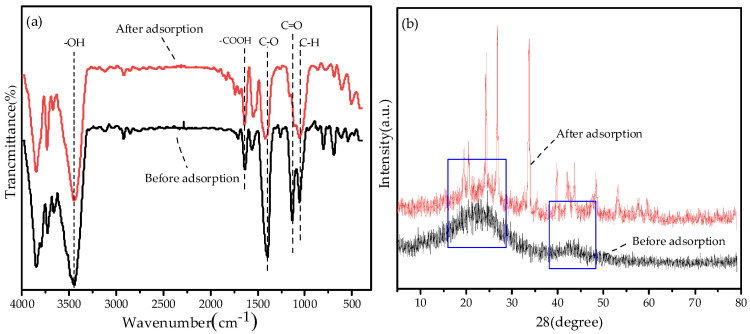
(**a**) The FTIR of biochar composites before and after adsorption; (**b**) The XRD of biochar composites before and after adsorption.

**Table 1 ijerph-19-04790-t001:** Design scheme of single-factor adsorption experiment.

Number	Factors	Variation Range
A	Solution pH	2, 3, 4, 5, 6
B	Rotating speed (rpm)	100, 125, 150, 175, 200
C	Adsorbent dosage (g)	0.005, 0.01, 0.02, 0.05, 0.1

**Table 2 ijerph-19-04790-t002:** The physical properties and element analysis of samples.

Samples	pH	Element Content (%)					Atomic Ratio			SSA (m^2^/g)	PV (cm^3^/g)	PD (nm)	Yield (%)	Ash (%)
		C	H	N	S	O ^a^	H/C	O/C	(O + N)/C					
B300	8.88	59.61	3.42	1.85	0.76	23.66	0.057	0.40	0.43	1.82	0.0122	1.753	43.8	10.70
B500	9.68	67.27	2.26	1.80	0.62	17.75	0.061	0.24	0.27	5.39	0.0151	1.773	33.5	10.30
B700	9.98	74.09	1.77	1.01	0.38	12.40	0.024	0.17	0.18	18.50	0.0166	1.851	26.6	10.35
BGO300	9.43	56.63	3.28	2.02	0.49	24.39	0.058	0.43	0.47	8.67	0.0155	1.789	45.3	13.19
BGO500	9.64	59.42	2.28	1.75	0.62	22.45	0.038	0.38	0.41	23.86	0.0173	1.991	29.4	13.48
BGO700	10.1	68.23	1.37	1.38	0.74	14.94	0.020	0.22	0.24	38.23	0.0189	2.114	27.4	11.34

“a” means that ω (O) was calculated by difference method; ω (O) = 100% − ω (C) − ω (H) − ω (N) − ω (S) − ω (ash). “SSA” means that specific surface area; “PV” means that pore volume; “PD” means that pore diameter.

**Table 3 ijerph-19-04790-t003:** The fitting parameters of isothermal adsorption model.

Model	Parameter	Temperature (°C)		
		25 °C	35 °C	45 °C
		$q_{e,\exp}\;=\;130.21$	$q_{e,\exp}\;=\;135.88$	$q_{e,\exp}\;=\;148.25$
Langmuir $q_{e}=\frac{K_{L}q_{\max}C_{e}}{1+K_{L}C_{e}}$	$q_{\max}$ (mg/g)	127.66	118.04	123.28
	$K_{L}$ (L/mg)	0.146	0.283	0.281
	$R^{2}$	0.838	0.848	0.744
	RSS	65.19	15.67	62.82
Freundlich $q_{e}=K_{F}C_{e}^{1/n}$	1/n	0.108	0.079	0.091
	$K_{F}$ (L/mg)	66.17	76.27	75.91
	$R^{2}$	0.970	0.945	0.894
	RSS	12.15	5.726	26.07
Temkin $q_{e}=B+{\mathrm{AlnC}}_{e}$	A	57.49	72.20	71.12
	B	11.48	8.268	9.709
	$R^{2}$	0.959	0.935	0.875
	RSS	16.47	6.689	30.63

$q_{e,\exp}$ is the actual experimental value (mg/g); $q_{\max}$
is the theoretical maximum adsorption (mg/g); $K_{L}$, 1/n, $K_{F}$, A, B are adsorption isotherm model equation parameters; R2 is the correlation coefficient; RSS is the sum of squares of residuals.

**Table 4 ijerph-19-04790-t004:** The thermodynamic parameters of Pb^2+^ adsorption by BGO700.

T(K)	ΔG (KJ/mol)	ΔH (KJ/mol)	ΔS (KJ/(mol·K)
298	−3.934	13.35	0.058
308	−4.514		
318	−5.094		

**Table 5 ijerph-19-04790-t005:** The fitting parameters of adsorption kinetics model.

**Kinetic Model**		**Quasi First Order Kinetics**				**Quasi Second Order Kinetics**			
		$\ln\left(q_{e}-q_{t}\right)\;=\;lnq_{e}-k_{1}t$				$\frac{t}{q_{t}}\;=\;\frac{1}{k_{2}q_{e}^{2}}+\frac{1}{q_{e}}t$			
Parameter	$q_{e,\exp}$	$q_{e,\mathrm{cal}}$	$k_{1}$	$R^{2}$	RSS	$q_{e,\mathrm{cal}}$	$k_{2}\left(\times 10^{-5}\right)$	$R^{2}$	RSS
BGO700	124.42	48.21	0.00127	0.974	0.028	128.53	6.73	0.999	0.026
B700	79.1	37.826	0.00144	0.949	0.036	81.9	9.54	0.998	0.205
**Kinetic Model**		**Intra-Particle Diffusion Model**							
		$q_{t}\;=\;K_{P}t^{0.5}+C$							
Parameter	$q_{e,\exp}$	$K_{P1}$	$C_{1}$	$R_{1}^{2}$	${\mathrm{RSS}}_{1}$	$K_{P2}$	$C_{2}$	$R_{2}^{2}$	${\mathrm{RSS}}_{2}$
BGO700	124.42	2.936	59.52	0.986	12.68	0.704	89.42	0.902	34.44
B700	79.1	1.453	39.63	0.983	1.818	0.508	53.85	0.928	8.036

$q_{e,\exp}$ is the actual experimental value (mg/g); $q_{e,\mathrm{cal}}$ is the theoretical value (mg/g); $k_{1},{\;k}_{2},{\;K}_{P1,2,}C_{1,2}$ are kinetic equation fitting parameters; R^2^ is the correlation coefficient; RSS is the sum of squares of residuals.

## Data Availability

The study did not report any data.

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
