# Peer review of "The Removal of Pb^2+^ from Aqueous Solution by Using Navel Orange Peel Biochar Supported Graphene Oxide: Characteristics, Response Surface Methodology, and Mechanism"

_ijerph, 2022, doi:10.3390/ijerph19084790_

Round 1

Reviewer 1 Report

The ms ID ijerph-1591315 deals with the preparation of an adsorbent composite material using biochar (from orange peel waste)-graphene oxide and evaluates its capacity for lead removal from aqueous solution, also considering the optimal adsorption parameters under different conditions of pH, rotating speed and amount of adsorbent. In addition, thermodynamic and kinetic models were used to describe the adsorption process and SEM-EDS, XPS, XRD and FTIR techniques were used to characterize and to describe Pb adsorption mechanisms in the composite material.

The Introduction clearly exposes the state of the art, mentioning relevant references. Material and Methods adequately described the several procedure steps of the experimental design, and the Results are adequately exposed and discussed. Although the importance of studying new and low-cost adsorbent materials, the study concerning the adsorption and desorption processes using biochar and heavy metals are not a great novelty.

Although I am not a native English and I don't feel qualified to judge about the English language and style, I leave below some suggestions to improve the reading of the ms:

L25-please use the same order for the optimal conditions described in L24, i.e., pH; adsorbent dosage and rotating speed.

L25-please change ‘is between’ by ‘ranged from 54.3 to 63.3%’.

L34-use ‘amount’ instead of ‘amounts’ and delete ‘were’.

L65 and 68, respectively: ‘Huang et al. (2017) [28] obtained…’ and ‘Chen et al (2013) [29] found…’

L90- ‘About of 73g of biomass and …’

L92- ‘It was then freeze-dried for…’

L93- ‘We adopted a slow heating and oxygen-limited pyrolysis conditions, heating to…’

L95- ‘It was used a N2 flow rate of …’

L104-‘ BGO composite was put into a…, and 50ml of Pb2+ solution (100 mg/L) was added.’

L105-106- please change to: ‘The experimental operating temperature was 25 ± 2 °C and the shaking speed was set to…’

L107- ‘The pH value was adjusted to …’

L110-111- ‘…the concentration of Pb2+ in each sample was tested after…’

L128- ‘, respectively,’ after ‘173 r/min.’

L130 and L139- ‘… the concentration of...was tested after…’

L141-please start the sentence with: ‘An amount of 0.086 g BGO…’

L143-144: ‘…was stirred in a water-bath constant temperature shaker for 24h and then…’ and ‘The saturated adsorbent was…’

L149- ‘…were dried in two different conditions, into the vacuum oven and under an ultraviolet light’.

L163- ‘…and the pore volume of the samples…’

L197- ‘The GO supported on biochar was…’

L199- ‘…the percentage of O element content increased (Table 2), which indicates …’

L208- ‘material showed also showed a peak at 2357.49 cm-1, where…’

L210-211: ‘Moreover, before 1700 cm-1 the composite material showed alkyl, aromatic and …’

L221-222: the phase is not clear, also to which condition it refers. Please rewrite it.

L223: replace ‘most scholars’ by ‘other studies’

L223-227: To render the text clearer for the reader, perhaps the sentence could be changed as: ‘In addition, orange peel biochar showed the worst adsorption capacity (B300…, B500…, B700…), which was significantly improved by about 97, 78 and 62% when biochar was supported by the GO (BGO300…, BGO500…, BGO700…). This indicates…’

L236: delete ‘conditions’

L237: change to: ‘4a, which indicated the pH 6 as the best condition for the adsorption of Pb2+.

L238-239: ‘…the adsorption capacity of Pb decreased, showing the lowest value at the pH 2.’

L240-242: ‘…, which will compete with Pb2+. In addition, the radius of hydrated H+ ion is smaller and easily adsorbed than the ion Pb2+, thus competing for the adsorption site of BGO and…’

L242-245: ‘With the increase of solution pH, H+ ion decreased and the negative charge on the BGO surface increased, together with the electrostatic attraction between Pb2+ and BGO charged sites.’

L346: ‘…by BGO700 is higher at the highest temperature.’

L347: ‘This is due to the enhanced thermal and diffusion movement of the molecules, which…’

L363: ‘…adsorption process is easy to take place and multi-layer adsorption can occur.

L384-386: the sentence is not easy to understand, please rewrite in a better way.

L431-440: The phrases are not clear to me. Moreover, the values of desorption efficiency described in L440 (around 50-60%, respect to which conditions?) seems in disagreement with the very low values of desorption capacity in the Figure 9b (about 3-4 mg/g). Please rewrite the paragraph.

Table and Figures:

L190 – Table 2: replace ‘sample’ by ‘samples’; replace ‘Scheme 2’ by ‘Samples’ and define ‘SSA’, ‘PV’ and ‘PD’.

L202- Figure 1 caption: SEM-EDS images (1) and component spectra (2) of the obtained materials: (a) biochar, (b) graphene oxide and (c) biochar-graphene oxide composite material. Yellow square in the images indicates the zone where the EDS spectra was performed.

L214-Figure 2 caption: FTIR spectra of the obtained materials: B=biochar, GO=graphene oxide, BGO=composite biochar-graphene oxide

L232- Figure 3 caption: Adsorption capacity of biochar (B), composite biochar-graphene oxide (BGO) and graphene oxide (GO) obtained at different pyrolyzed temperatures.

Author Response

Dear Reviewer:

Thank you for your comments concerning our manuscript entitled ' The Removal of Pb2+ from Aqueous Solution by Using Navel Orange Peel Biochar Supported Graphene Oxide: Characteristics, Response Surface Methodology and Mechanism' (Manuscript Number: ijerph-1591319). These comments are valuable and helpful for revising and improving our paper, as well as the important guiding significance to our researches. We have studied all the comments carefully, and the revised parts of manuscript were marked in yellow. The point-to-point responses to the reviewers’ comments are as following:

Point 1: L221-222: the phase is not clear, also to which condition it refers. Please rewrite it.

Response 1: We are sorry for our incomplete writing, we have rewritten it. (P6, Lines 229-236). The specific modification is as follow:

The adsorption capacity of BGO700 was increased by 134.9% compared to BGO300, and B700 was increased by 187.1% compared to B300. This is due to the influence of pyrolysis temperature, which is consistent with the results of other studies [39-40]. In addition, orange peel biochar showed the worst adsorption capacity (B300: 21.7 mg/g; B500: 43.43 mg/g; B700: 62.3 mg/g), which was significantly improved by about 97%, 78 % and 62% when biochar was supported by the GO (BGO300: 42.82 mg/g; BGO500: 77.46 mg/g; BGO700: 100.61 mg/g).

Point 2: L384-386: the sentence is not easy to understand, please rewrite in a better way.

Response 2: According to reviewer’s comment, we have rewritten the sentence, (P11, Lines 360-365)the specific modifications are as follow:

The adsorption thermodynamic equation and parameters were used to describe the state characteristics of the adsorption process. According to the experimental results of the isotherm adsorption of Pb2+ by BGO700, the linear equation of ln(KL)-1/T was fitted with 1/T as the abscissa and ln(KL) as the ordinate. The parameter fitting results were shown in Figure 7, and the correlation coefficient R2=0.989, which has a better linear relationship.

Point 3: L431-440: The phrases are not clear to me. Moreover, the values of desorption efficiency described in L440 (around 50-60%, respect to which conditions?) seems in disagreement with the very low values of desorption capacity in the Figure 9b (about 3-4 mg/g). Please rewrite the paragraph.

Response: When looking up references, some studies found that UV can introduce oxygen-containing functional groups on the surface of biochar, which can increase the content of surface functional groups. At the same time, UV is also a low-cost method. So In this study, a small experiment was carried out to provide a preliminary basis for subsequent experiments. the values of desorption efficiency = desorption capacity/(adsorption capacity * Madsorbent)*100%. According to reviewer’s comment, we have rewritten the paragraph, (P11, Lines 360-365)the specific modifications are as follow:

The adsorption capacity of UV/BGO700, bake/BGO700, UV/B700, and bake/B700 gradually were decreased, and finally reached a stable trend, which were 57.4 mg/g, 50.6 mg/g, 26.6 mg/g and 28.5 mg/g, respectively. The reason for the low adsorption efficiency may be that the Pb2+ through the action of hydrogen bonds, π-π bonds and oxygen-containing functional groups was not completely desorbed by HCl. In addition, the adsorption capacity of the sample irradiated by ultraviolet light is slightly higher than that of the vacuum-dried sample, which is due to the introduction of some oxygen-containing functional groups on the surface of the sample by ultraviolet light irradiation [48]. According to figure 9(b), the desorption efficiency ranged from 54.3 to 63.3%.

Thank you again for your suggestions for this paper.

Reviewer 2 Report

This work entitled “The Removal of Pb2+ from Aqueous Solution by Using Navel Orange Peel Biochar Supported Graphene Oxide: Characteristics, Response Surface Methodology and Mechanism” by Liu et al. prepared biochar supported graphene oxide (BGO) sorbents and tested their adsorption performance on Pb2+ removal from water. The working conditions were optimized and mechanisms were elucidated by using batch experiments and multiple characterization techniques. This topic is of potential interest to the readers of the International Journal of Environmental Research and Public Health. I suggest the authors consider the comments below and revise this manuscript accordingly.

  1. The manuscript is not written well and needs tremendous work on language editing and refining. For example, Lines 21-22. Please also rephrase the sentences to avoid plagiarism (e.g., Lines 45-46, Lines 160-163, and more). Revise the manuscript thoroughly and carefully.
  2. Write out a full name when an abbreviation is given. For example, Lines 20-21 and more.
  3. Introduction Paragraphs 2 and 3 are too general information about biochar and GO. The authors should write logically to show the readers why you modify biochar with graphene. In another way, what is the Rationale and Significance for this study?
  4. Many experimental conditions are not clearly presented. For example, the fitting models should be introduced in Section 2.
  5. Table 2. Add footnotes for SSA, PV, and PD. Why there is no characterization data for GO300, 500, and 700.
  6. Figure 2. Figure legend should be B700, GO700, and BGO700, right?
  7. Lines 227-229: GO is the best one among biochar, biochar/GO, and GO. Why you still prepare biochar/GO composites? Please refer to comment #3.
  8. Lines 243-244: please add zeta potentials of BGO under different pH conditions.
  9. Section 3.3 about Response Surface Methodology. I recommend shortening this part and moving it to supporting information. Because this is not the focus of this work, the readers only care about the main conclusions, that is the optimal condition.
  10. Figure 6a. the adsorption isotherms are more linear-shaped. Any thoughts and explanations?
  11. Section 3.6. please explain why pick UV and bake for regenerating the sorbents? The results showed that desorption efficiency is not high 54.3%‒3%. any other strategies for desorption?
  12. Figure 11. A better caption is needed.
  13. Lines 488-489. Does this hypothesis contradict the MINTEQ calculation (Figure 4b) that “Pb exists in ionic state” under pH 6?

Reviewer 3 Report

Manuscript talks about how graphene oxide from orange peel can be used to adsorb Pb2+, other transition metals, and cations in wastewater.

This manuscript goes through a plethora of measurement methods and very detailed in giving results, BUT the narrative of the manuscript needs significant improvement.

Abstract is weak and confusing, especially the latter half of it. For example, line 22 is missing half of the sentence that describe the difference in the materials that give different percentages

Introduction is very jumpy, and needs a better work in establishing bridges between different paragraph. For example, the introduction of GO that came out of nowhere, while it should have been prefaced by one of the most common carbon-based materials is graphene oxide... Or the justification on why using graphene oxide, why not graphite oxide or carbon nanotube/nanowire/coal/etc

Methods are not justified, which leads me to believe that although extensive, half of the data in the discussion is missing. For example: why 1:1.2:6.46 ratio of flake:KNO3:KMnO44:H2SO4, why 20 mL/min flow rate of N2, why 0.02 g of BGO composite that is added into a 100 mL polyethylene centrifuge tube? How do the authors chose these numbers specifically? and the reasons behind why the authors decide on these numbers
i.e. what happens if i use 10 or 50 mL/min flow rate for N2, etc

What disturbs me the most is the optimal values the authors presented in the abstract section do not match their variation range in Table 1. In abstract: "The optimal conditions were as follows: solu-24 tion pH was 4.97, rotating speed was 172.97 r/min, and adsorbent dosage was 0.086 g." 
while in Table 1, pH was increased every 1 unit, rotating speed every 25, and dosage dont have 0.086 data point.

Author Response

Dear Reviewer:

Thank you for your comments concerning our manuscript entitled ' The Removal of Pb2+ from Aqueous Solution by Using Navel Orange Peel Biochar Supported Graphene Oxide: Characteristics, Response Surface Methodology and Mechanism' (Manuscript Number: ijerph-1591319). These comments are valuable and helpful for revising and improving our paper, as well as the important guiding significance to our researches. We have studied all the comments carefully, and the revised parts of manuscript were marked in yellow. The point-to-point responses to the reviewers’ comments are as following:

Point 1: Abstract is weak and confusing, especially the latter half of it. For example, line 22 is missing half of the sentence that describe the difference in the materials that give different percentages.

Response 1: According to reviewer’s comment, we have added more describe of the abstract, the specific modifications are as follow:

The optimal adsorption parameters were analyzed by response surface methodology under the conditions of solution pH, adsorbent dosage and rotating speed. The adsorption isotherm and kinetic model fitting experiments were carried out according to the optimal adsorption parameters, and the mechanism of BGO adsorption of Pb2+ was explained using Scanning Electron Microscope (SEM-EDS), X-ray Photoelectron Spectroscopy (XPS), X-ray Diffraction (XRD), Fourier Transform Infrared Spectroscopy (FTIR). Compared with virgin biochar, the adsorption capacity of Pb2+ on biochar supported graphene oxide was significantly increased.

Point 2: Introduction is very jumpy, and needs a better work in establishing bridges between different paragraph. For example, the introduction of GO that came out of nowhere, while it should have been prefaced by one of the most common carbon-based materials is graphene oxide... Or the justification on why using graphene oxide, why not graphite oxide or carbon nanotube/nanowire/coal/etc?

Response 2: According to reviewer’s comment, we have added the information in revised paper , the specific modifications are as follow:

TheBiochar can be understood as a continuum of pyrolysis products with very complex chemical composition and characteristics [19]. It contains a variety of negatively charged functional groups [20-21], which can fix heavy metal ions on the surface of biochar. However, there are some problems in the virgin biochar materials, such as limited specific surface area, low porosity and few adsorption sites, which limit its wide application[22]. Therefore, studies enhance the adsorption capacity of biochar by modifying and loading functional materials [23-24]. Graphene oxide (GO) is the main derivative of graphene. It has attracted wide attention because of its large specific surface area and abundant oxygen-containing functional groups[25-26]. Huang et al. (2017) [27] found that the pore volume, specific surface area and adsorption performance of BGO were significantly increased. Chen et al. (2013) [28] found that graphene oxide chitosan composite material has a high treatment capacity for heavy metal pollutants.

 There are abundant navel orange resources in South Jiangxi, which are deeply loved by people. However, the navel orange peels has low economic value and is often discarded, which releases a large amount of CO2 and causes environmental pollution[29]. The navel orange peel contains abundant plant fibers and functional groups, which is a good raw material for preparing biochar. Therefore, this study selected lead, a common and typical heavy metal pollutant in the water environment, as the adsorption object, navel orange peel and flake graphite were used as the raw materials to prepare BGO composites. The optimal adsorption parameters under solution pH, rotating speed and dosage were analyzed by response surface method. Thermodynamic and kinetic models were used to describe the adsorption process of BGO composites. Combined with Scanning Electron Microscope (SEM-EDS), X-ray Photoelectron Spectroscopy (XPS), X-ray Diffraction (XRD), Fourier Transform Infrared Spectroscopy (FTIR), the adsorption mechanism of BGO for Pb2+ in aqueous solution was described. The results will provide some basic experimental data and material information for the effective treatment of heavy metals in sewage, and provide new ideas for the resource reuse of navel orange peel.

Point 3:  Methods are not justified, which leads me to believe that although extensive, half of the data in the discussion is missing. For example: why 1:1.2:6.46 ratio of flake:KNO3:KMnO4:H2SO4, why 20 mL/min flow rate of N2, why 0.02 g of BGO composite that is added into a 100 mL polyethylene centrifuge tube? How do the authors chose these numbers specifically? and the reasons behind why the authors decide on these numbers, i.e. what happens if i use 10 or 50 mL/min flow rate for N2, etc?

Response 3: we are sorry for our negligence of the statement of the methods,The graphene oxide was prepared by the modified hummer method, the ratio of flake graphite: KNO3: KMnO4: H2SO4 was 1:1.2:6:46. The preparation of graphene oxide material was completed under the guidance of Ningbo Institute of Materials Technology and Engineering, Chinese Academy of Science. The ratio of graphene oxide prepared  according  to zhou et al[1]。0.02 g of BGO composite was added to In order to achieve adsorption equilibrium. 20 mL/min flow rate of N2 to isolate oxygen, if the flow rate is too small, it is difficult to achieve the goal. On the contrary, if the flow rate is too large, it is easy to blow off the finished biochar.

  • Zhou X., Liu Z. A Scalable, Solution-Phase Processing Route to Graphene Oxide and Graphene Ultralarge Sheets. Chem Comm 2010, 46:2611-2613.

Point 4:  What disturbs me the most is the optimal values the authors presented in the abstract section do not match their variation range in Table 1. In abstract: "The optimal conditions were as follows: solu-24 tion pH was 4.97, rotating speed was 172.97 r/min, and adsorbent dosage was 0.086 g." while in Table 1, pH was increased every 1 unit, rotating speed every 25, and dosage dont have 0.086 data point.

Response 4: The single factor effect experiment was the first to be discussed prior to the optimal conditions, in which pH was increased every 1 unit, rotating speed every 25, and dosage was increased every 0.005, 0.01, 0.02 g... Single factor experiment and response surface experiment are independent. In the actual environment, it is not a single factor that affects the adsorption of heavy metals by biochar, but a combination of multiple factors. Therefore, the optimal values of the response surface experiment were selected. Experimental design was shown in Table 1. The optimal The conditions results are based on the Box-Behnken response surface.

Table 1. Experimental design and results.

number

Experimental factors

adsorption capacity (mg/g)

A: solution pH

B: rotating speed (r/min)

C: adsorbent dosage (g)

1

6

150

0.005

81.35

2

4

100

0.005

75.7

3

2

100

0.0525

2.27

4

2

200

0.0525

2.77

5

4

150

0.0525

109.49

6

6

150

0.10

103.66

7

4

200

0.10

117.12

8

4

100

0.10

97.64

9

2

150

0.10

1.985

10

4

150

0.0525

110.0

11

4

150

0.0525

109.48

12

2

150

0.005

1.57

13

4

150

0.0525

108.96

14

6

100

0.0525

98.16

15

4

200

0.01

84.6

16

6

200

0.05

100.32

17

4

150

0.05

109.68

Thank you again for your suggestions for this paper.

Round 2

Reviewer 2 Report

The authors have fully addressed my comments. The revised manuscript now can be accepted for publication.

Author Response

Thanks again for your comments concerning our manuscrip. Have a nice day!

Reviewer 3 Report

This manuscript is a huge improvement from the previous version. However, I still have a few points that I would like clarifications on. My main issue is still with the experimental section, which I think need to have a clearer narrative. The specific points of confusion are listed below, along with other comments and questions:

page 2 line  51-52: please add the following references for oxygen surface functionalization and its effect on reaction rates:1) J. Chem. Phys. 155, 134702 (2021), 2) https://doi.org/10.26434/chemrxiv-2021-5gv6w, 3) ACS Appl. Mater. Interfaces 2021, 13, 7, 8169–8180

page 2 line 92: why 50 ml Pb2+? what’s the justification? why not higher or lower? what’s the concentration of the Pb2+ in this solution?

page 3 line 94 : why shaking speed was set to 150 rpm? Also throughout the manuscript: should be rpm, not r/m

page 4 line 99, and line 119: no point of determining the Pb2+ concentration after experiments if the initial concentration are not known

page 4 line 105: central composite design need to be explained. And what are low, medium and high levels of this experiment? also needs to be explained

page 4 line 111: also need to explain in text: single-molecular-layer Langmuir model, multi-molecular-layer Freundlich model and diffusion interface Temkin model (this were given on page 10 line 334-335 for Langmuir, and 345-348 for the other two, but should also be given up front after these models were first introduced in on page 4 line 111)

page 4 line 116-118: no explanation of the optimal experiment conditions. And if the speed were manually adjusted every 25 increments in accordance to Table 1, where the 173 rpm came from?

page 4 line 121: there was no definition of B700 prior to this point. Also what happened to the BGO300 and BGO500? Why authors prepared them and then didn’t include them in their experiments? The omission of BGO300 and BGO500 is the most confusing part of the manuscript. I really think an extra table that gives the all the sample naming and the condition they are in (B/GO/BGO 300/500/700/ UV/bake) along with all the variables of the conditions the sample were subjected in (range of adsorption temperatures:25, 35, 45, 0.5 mol/L Cl, xx-200 rpm optimum rotation speed, etc) would help the understanding of the manuscript tremendously and to help keep easier track with the narrative in the discussion section. Something along with The Table 1 authors provided to the comments in the previous round on experimental design and results would be helpful to have in the main body of the manuscript

page 4 line 130: which BGO? raw/300/500/700? need to clarify

page 4 line 134: need to justify why authors chose 0.5 mol/L concentration of HCl solution

page 5 line 195: the B/GO needs to be defined, and the differences with BGO700 need to be clarified in the method section prior to discussion

page 5 line 206-208: Caption needs to be consistent with text – use B/GO/BGO instead of full spelling. Also what happened to the 300 and 500 samples?

page 6 line 216-218: “Moreover, before 1700 cm-1 the composite material showed alkyl, aromatic and some oxygen-containing groups. These functional groups can provide adsorption sites and enhance the adsorption performance of the composite material” contradicts line 186-191: “which indicates that the higher degree of carbonization and more 188 complete π-conjugated aromatic structure with high pyrolysis temperature… The specific surface area was increased, which indicates that the pyrolysis temperature is the key factor affecting the specific surface area.” If I misunderstood, then a clearer narrative should be given to avoid confusion, whether authors seek to have more oxygen functionals or prefer to have a pristine surface

Page 7 line 240 and also figure 4: optimal pH is 6, which contradicts line 26 that stated optimal pH is 4.97.

Page 7 line 254-257: the sentence contradicts itself, and the earlier part of the paragraph: formation of OH complex reduce the charge of Pb2+, which is more conducive to the adsorption of Pb2+? The earlier part of this paragraph said that high pH with hydroxyl will lead to Pb(OH)2 precipitation – which is at all not conducive for Pb2+ adsorption

Page 7 line 260-261: “. It can be seen from the figure that the efficiency of Pb2+ adsorption was slightly reduced under low and high-speed conditions” – the effect of high speed rotation was explained, but not the low speed. Also in this paragraph, the optimum range of rotation should be given (i.e. low part – up 200 rpm) – which can also define the “central speed of rotation” given in line 296-297 on page 9

Page 9 Figure 5: Figure a-c are not well descried in the manuscript. Authors should add a few sentences on the paragraph in line 290-307 somewhere along the line of: in fig 5 we compare the effect of rotating speed and adsorbent dosage against solution pH. as can be seen from fig c, the flat curvature shows the effect of rotating speed and adsorbent dosage are negligible in comparison to the solution pH

Page 9 line 302-305: “Therefore, there is no linear relationship between the three factors from the results, the order of the  degree of influence on the adsorption of Pb2+ was solution pH>adsorbent dosage>rotation speed.” – there is no way to tell from Fig 5 that: 1) there is no linear relationship and 2) the degree of influence. More explanation/ guidance required on how the authors came to these conclusions

Page 9 line 305-308: the results of quadratic multiple regression that shows the optimum pH is 4.97 needs to be shown in the manuscript. Especially since in the previous paragraphs authors made a big discussion why pH 6 was the best condition

Author Response

Please see the attachment,thank you.
